# Maximum Entropy On-Policy Actor-Critic via Entropy Advantage Estimation

## Abstract

Entropy regularisation is a widely adopted technique that enhances policy optimisation performance and stability. Many practical on-policy methods employ an entropy regularisation term to the policy gradient, thereby maximising policy entropy at visited states. On the other hand, another form of entropy regularisation, maximum entropy reinforcement learning (MaxEnt RL), augments the standard objective with an entropy term, aiming to maximise both the cumulative reward and the entropy of the trajectories induced by a policy. However, despite its empirical and theoretical achievements, its application in on-policy actor-critic contexts remains relatively underexplored. In this work, we propose an on-policy actor-critic algorithm based on the MaxEnt RL framework. A key aspect of our approach is separating the entropy objective from the MaxEnt RL objective. This delineation allows us to introduce an additional critic for the entropy objective alongside the conventional value critic. It also offers finer control over the optimisation process, incorporating a discount factor specifically for the entropy that provides a distinct way to balance the original and entropy objectives. Our empirical evaluations demonstrate that extending Proximal Policy Optimisation (PPO) and replacing its entropy regularisation with the proposed method significantly improves the performance of PPO in both continuous control tasks and across 16 Procgen environments. Additionally, the results underline MaxEnt RL's capacity to enhance generalisation.

## 1 Introduction

Reinforcement learning (RL) has seen significant success across diverse domains like robotics (Peng et al., 2018; Rudin et al., 2022), games (Vinyals et al., 2019; Berner et al., 2019), and training language models (Ouyang et al., 2022). Central to these achievements are on-policy actor-critic algorithms such as Proximal Policy Optimisation (PPO) (Schulman et al., 2017b) and IMPALA (Espeholt et al., 2018).

Entropy Regularisation (ER) is a popular technique to improve such on-policy optimisation methods, posited to encourage exploration and prevent a policy from being prematurely deterministic (Ahmed et al., 2019). A prevalent approach of ER in practical policy gradient algorithms incorporates the inclusion of the entropy term to the sample-based policy gradient estimator to maximise the policy entropy at each sampled state, retaining the stochasticity of the policy during the optimisation process (Williams & Peng, 1991; Mnih et al., 2016; Schulman et al., 2017b). Despite its empirical success, this method remains a heuristic approach without solid theoretical understanding (Ahmed et al., 2019).

On the other hand, Maximum Entropy Reinforcement Learning (MaxEnt RL) introduces an entropy regularisation term to the conventional RL objective, aiming to optimise cumulative rewards and the trajectory entropy induced by a policy. While the ER method mentioned above seeks to maximise policy entropy at visited states, MaxEnt RL directs a policy toward regions of higher expected trajectory entropy, albeit at the cost of bias imposed on the objective (Levine, 2018).

Recent studies have shown several compelling theoretical properties of MaxEnt RL, such as robustness to external perturbation (Eysenbach & Levine, 2021; 2019) and provably efficient exploration (Hazan et al., 2019; Tiapkin et al., 2023). Notably, (Mei et al., 2020) established that MaxEnt RL

can accelerate the convergence of the policy gradient method under softmax parameterisation in a tabular setting.

Despite the theoretical studies, the implementation of MaxEnt RL has been dominated by off-policy variants. Importantly, Soft Actor-Critic (Haarnoja et al., 2018), an off-policy MaxEnt algorithm, has become a standard approach in continuous control tasks (Ball & Roberts, 2021). This lack of examination of practical on-policy MaxEnt RL algorithms restricts the broader applicability of MaxEnt RL, especially in scenarios where off-policy algorithms might falter. For instance, their tendency for data overfitting compromises generalisation to unseen environments (Zhang et al., 2018).

To this end, we introduce Entropy Advantage Policy Optimisation (EAPO), a model-free on-policy Actor-Critic algorithm based on the MaxEnt framework. The central idea is to independently estimate the conventional and entropy advantages by performing the Generalised Advantage Estimation (GAE) (Schulman et al., 2015b), utilising a distinct critic for each objective.

The separated advantage estimation provides us with the means to alleviate the difficulty of learning the MaxEnt RL objective:

1. It enables us to control the variance-bias tradeoff of each estimation independently.
2. It allows each function approximator to learn each target of the different scales in isolation instead of tracking the combined objective.
3. Introducing a distinct discount factor for entropy estimate mitigates the problem of favouring a long trajectory in MaxEnt RL (Yu et al., 2022b) by reducing the effective horizon.

EAPO requires only minor modifications to existing advantage actor-critic algorithms to estimate the trajectory entropy using sampled log probabilities. Furthermore, it mandates minimal overheads by utilising parameter sharing among critics.

We extend the well-established PPO to implement EAPO and evaluate it in 4 MuJoCo locomotion tasks (Todorov et al., 2012) and 16 Procgen benchmark environments (Cobbe et al., 2020). Our empirical results demonstrate that EAPO outperforms the previous ER method across environments with both discrete and continuous action spaces. Further, we show the improved generalisation capability that supports the claimed robustness of MaxEnt RL algorithms (Eysenbach & Levine, 2021).

Therefore, given its simplicity and effectiveness, this study provides a versatile tool for applying MaxEnt RL across diverse settings. The scenarios include areas where on-policy algorithms excel, such as exploration (Hazan et al., 2019), multi-agent scenarios (Yu et al., 2022a), and Markov games (Cen et al., 2022b).

## 2 BACKGROUND

### 2.1 PRELIMINARIES

This work considers a finite discounted Markov Decision Process $\langle \mathcal{S}, \mathcal{A}, P, r, \rho, \gamma_V, \gamma_{\mathcal{H}} \rangle$, where $\mathcal{S}$ is the set of states $s$ and $\mathcal{A}$ is the set of actions $a$, and $\rho$ is the initial state distribution. $P$ is the transition function $P : \mathcal{S} \times \mathcal{A} \times \mathcal{S} \mapsto [0, 1]$, and $r$ is the reward function $r : \mathcal{S} \times \mathcal{A} \mapsto \mathbb{R}$. $\gamma_V$ and $\gamma_{\mathcal{H}}$ are the discount factors. We define the value of state $s$ under the policy $\pi$ as

$$V^\pi(s) := \mathbb{E}_{\substack{s_0=s, a_t \sim \pi(\cdot|s_t), \\ s_{t+1} \sim p(\cdot|s_t, a_t)}} \left[ \sum_{t=0}^{\infty} \gamma_V^t r(s_t, a_t) \right]. \tag{1}$$

Also, the action-value of performing action $a$ at state $s$ under the policy $\pi$ is

$$Q^\pi(s, a) := \mathbb{E}_{\substack{s_0=s, a_0=a, \\ a_{t>0} \sim \pi(\cdot|s_t), \\ s_{t+1} \sim p(\cdot|s_t, a_t)}} \left[ \sum_{t=0}^{\infty} \gamma_V^t r(s_t, a_t) \right]. \tag{2}$$

And define the advantage function $A^\pi$ as

$$A^\pi(s, a) := Q^\pi(s, a) - \mathbb{E}_{\pi(\cdot|s)} \left[ Q^\pi(s, \cdot) \right] \tag{3}$$

$$= Q^\pi(s, a) - V^\pi(s). \tag{4}$$

We also define the discounted policy-induced trajectory entropy or the entropy rate of state $s$ under policy $\pi$ as

$$V_{\mathcal{H}}^{\pi}(s) := \mathbb{E}_{\substack{s_0=s, a_t \sim \pi(\cdot|s_t), \\ s_{t+1} \sim p(\cdot|s_t, a_t)}} \left[ \sum_{t=0}^{\infty} -\gamma_{\mathcal{H}}^t \log \pi(a_t|s_t) \right]. \tag{5}$$

This trajectory entropy is the Shannon entropy of the distribution of possible future trajectories of the MDP with deterministic dynamics (Levine, 2018; Tiapkin et al., 2023). The goal of Maximum Entropy Reinforcement Learning (MaxEnt RL), or often Regularised MDPs (Geist et al., 2019; Neu et al., 2017) is to maximise the expectation of the sum of the value and the trajectory entropy with respect to the initial state distribution:

$$J(\pi) = \mathbb{E}_{\substack{s_0 \sim \rho, \\ a_t \sim \pi, \\ s_{t+1} \sim p}} \left[ \sum_{t=0}^{\infty} \gamma_V^t r(s_t, a_t) - \gamma_{\mathcal{H}}^t \tau \log \pi(a_t|s_t) \right] \tag{6}$$

$$= \mathbb{E}_{s_0 \sim \rho} \left[ V^{\pi}(s_0) + \tau V_{\mathcal{H}}^{\pi}(s_0) \right] \tag{7}$$

where the temperature parameter $\tau \geq 0$ is a hyperparameter to be controlled to balance the significance between these two objectives, and we introduce the distinct discount factors.

## 2.2 SOFT ADVANTAGE FUNCTION

Analogous to the definition of the action-value function $Q^{\pi}$ as the expected cumulative rewards after selecting an action $a$ (Sutton & Barto, 2018), we define $Q_{\mathcal{H}}^{\pi}$ as the expected future trajectory entropy after selecting an action:

$$Q_{\mathcal{H}}^{\pi}(s, a) := \gamma_{\mathcal{H}} \mathbb{E}_{s' \sim p(\cdot|s, a)} \left[ V_{\mathcal{H}}^{\pi}(s') \right]. \tag{8}$$

The definition arises naturally from the consideration that uncertainty exists due to the stochastic policy at the current state, which has settled by the time an action is performed. Consequently, the expected discounted future trajectory entropy is simply the expectation over trajectory entropies of the possible subsequent states.

From the recursive relation of trajectory entropy from (5) and the defintion (8), the following relation is derived:

$$V_{\mathcal{H}}^{\pi}(s_t) = \mathbb{E}_{a_t \sim \pi(\cdot|s_t)} \left[ -\log \pi(a_t|s_t) + Q_{\mathcal{H}}^{\pi}(s_t, a_t) \right]. \tag{9}$$

We now define the entropy advantage function $A_{\mathcal{H}}^{\pi}$ analogous to (3):

$$A_{\mathcal{H}}^{\pi}(s, a) := Q_{\mathcal{H}}^{\pi}(s, a) - \mathbb{E}_{\pi(\cdot|s)} \left[ Q_{\mathcal{H}}^{\pi}(s, \cdot) \right] \tag{10}$$

$$= Q_{\mathcal{H}}^{\pi}(s, a) - V_{\mathcal{H}}^{\pi}(s) + H(\pi(\cdot|s)), \tag{11}$$

where $H(\pi(\cdot|s))$ is the Shannon entropy of the policy at state $s$.

We let $\tilde{V}^{\pi}(s) := V^{\pi}(s) + \tau V_{\mathcal{H}}^{\pi}(s)$ as the soft value function, and let $\tilde{Q}^{\pi}(s, a) := Q^{\pi}(s) + \tau Q_{\mathcal{H}}^{\pi}(s, a)$ as the soft Q-function. Finally, we define the soft advantage function:

$$\tilde{A}_t^{\pi}(s, a) := A^{\pi}(s, a) + \tau A_{\mathcal{H}}^{\pi}(s, a) \tag{12}$$

$$= Q^{\pi}(s, a) - V^{\pi}(s) + \tau(Q_{\mathcal{H}}^{\pi}(s, a) - V_{\mathcal{H}}^{\pi}(s) + H(\pi))$$

$$= \tilde{Q}^{\pi}(s, a) - \tilde{V}^{\pi}(s) + \tau H(\pi(\cdot|s)). \tag{13}$$

## 2.3 SOFT POLICY GRADIENT THEOREM

The soft advantage function is then used for optimising the policy to maximise the MaxEnt RL objective.

**Theorem 1** (Soft Policy Gradient). *Let $J(\pi)$ the MaxEnt RL objective defined in 6. And $\pi_{\theta}(a|s)$ be a parameterised policy. Then,*

$$\nabla_{\theta} J(\pi_{\theta}) = \mathbb{E}_{\substack{s_0 \sim \rho, \\ a_t \sim \pi, \\ s_{t+1} \sim p}} \left[ \nabla_{\theta} \log \pi_{\theta}(a_t|s_t) \tilde{A}_t^{\pi}(s_t, a_t) \right]. \tag{14}$$

We provide the proof in Appendix A. The soft policy gradient theorem implies the MaxEnt RL objective can be optimised with a direct policy gradient method. Moreover, (Mei et al., 2020) established that the soft policy gradient has the global convergence property and may converge faster than the policy gradient without entropy regularisation despite the objective being biased.

## 3 RELATED WORKS

One of the most prominent aspects of the MaxEnt RL formulation that has been studied is the ability to connect policy gradient methods and off-policy value-based methods that learn the soft $Q$-function (Haarnoja et al., 2018; Nachum et al., 2017; O'Donoghue et al., 2016; Schulman et al., 2017a). However, this work focuses on the soft policy gradient using the soft advantage estimation in an on-policy setting.

Shi et al. (2019) explored the soft policy gradient method, emphasising its inherent simplicity. While they employed the soft $Q$-function to guide the policy gradient, linking their method to off-policy techniques, EAPO leverages the variance-reduced estimation of the entropy advantage function to formulate the soft advantage function suitable for on-policy algorithms. Moreover, Shi et al. (2019) introduced additional techniques to mitigate the challenging task of estimating the soft $Q$-function. However, EAPO can seamlessly integrate with existing techniques, such as value function normalisation, due to its structural equivalence between its method for estimating the entropy advantage function and the conventional advantage function.

Recent studies have investigated the theoretical properties of policy gradient methods (Agarwal et al., 2021; Mei et al., 2020; Cen et al., 2022a). Notably, the authors have shown that combining Natural Policy Gradient (NPG) methods (Kakade, 2001) and the entropy-regularised MDPs can speed up the convergence. Although some authors (Khodadadian et al., 2021; Shani et al., 2020) have drawn the connection between PPO and NPG methods rigorously in theoretical settings, it remains unclear whether EAPO, which extends PPO to the Regularised MDP setting, can benefit from the theoretical guarantees.

## 4 PROPOSED METHOD

### 4.1 OVERVIEW

In this section, we develop our Entropy Advantage Policy Optimisation (EAPO) algorithm. At its core, EAPO independently estimates both the value advantage function and the entropy advantage function and combines them to derive the soft advantage function. Alongside the conventional critic, EAPO adopts a separate approximator to predict the trajectory entropy of a state, which is then used for entropy advantage estimation. We extend the PPO (Schulman et al., 2017b) by substituting the advantage estimate with the soft advantage and omitting the entropy bonus term.

### 4.2 ENTROPY ADVANTAGE ESTIMATION

The entropy advantage $A_{\mathcal{H}}^{\pi}$ is estimated from the sampled log probabilities of the behaviour policy. We utilise the Generalised Advantage Estimation (GAE) (Schulman et al., 2015b) for a variance-reduced estimation of the entropy advantage:

$$\hat{A}_t^{\mathcal{H},\text{GAE}(\lambda_{\mathcal{H}},\gamma_{\mathcal{H}})} := \sum_{t=0}^{\infty} (\lambda_{\mathcal{H}}\gamma_{\mathcal{H}})^t \delta_{t+1}^{\mathcal{H}}, \tag{15}$$

where $\delta_t^{\mathcal{H}} := -\log \pi(a_t|s_t) + \gamma_{\mathcal{H}} V_{\mathcal{H}}^{\pi}(s_{t+1}) - V_{\mathcal{H}}^{\pi}(s_t)$, and $\gamma_{\mathcal{H}}$ and $\lambda_{\mathcal{H}}$ are the discount factor and GAE lambda for entropy advantage estimation, respectively. Note that the equation is the same as the GAE for the conventional advantage, except the reward term is replaced by the negative log probability. This simplicity is also consistent with the remark that the only modification required for the MaxEnt policy gradient is to add the negative log probability term to the reward at each time step (Levine, 2018).

### 4.3 ENTROPY CRITIC

A dedicated entropy critic network, parameterised by $\omega$, approximates the trajectory entropy $V_{\mathcal{H}}^{\pi}$, and it is trained by minimising the mean squared error:

$$L_t^{\mathcal{H}}(\omega) := \hat{\mathbb{E}}_t \left[ \frac{1}{2} \left( V_{\mathcal{H}}^{\pi}(s_t; \omega) - \hat{V}_{\mathcal{H}}^{\pi}(s_t) \right)^2 \right], \tag{16}$$

where the trajectory entropy estimate $\hat{V}_{\mathcal{H}}^{\pi}(s_t)$ is the sum of the entropy advantage estimate and the approximated trajectory entropy, i.e., $\hat{V}_{\mathcal{H}}^{\pi}(s_t) = \hat{A}_t^{\mathcal{H},\text{GAE}(\lambda_{\mathcal{H}},\gamma_{\mathcal{H}})}(s_t, a_t; \omega) + V_{\mathcal{H}}^{\pi}(s_t; \omega)$. Throughout the conducted experiments, we implemented the entropy critic network to share its parameters with the return value critic $V_\phi^V$, with only the final linear layers for outputting its prediction distinct. This form of parameter sharing allows minimal computational overhead to implement EAPO.

Further, we employ the PopArt normalisation (van Hasselt et al., 2016) to address the scale difference of entropy and return estimates. It is important to note that the negative log probability $-\log \pi(a_t|s_t)$ is collected for every timestep. In contrast, the reward can be sparse, leading to significant magnitude variations based on the dynamics of the environment (Hessel et al., 2019). This discrepancy can pose challenges, especially when using a shared architecture. Thus, utilising the value normalisation technique like PopArt is pivotal for the practical implementation of EAPO.

## 4.4 Entropy Advantage Policy Optimisation

Subsequently, we integrate the entropy advantage with the standard advantage estimate $\hat{A}_V^{\pi}$, also computed using GAE and return value critic parameterised by $\phi$ analogously to the entropy advantage estimation process we describe above. Then the soft advantage function $\hat{A}_t^J$ is

$$\tilde{A}_t^{\pi} = \hat{A}_t^{V,\text{GAE}(\lambda_V,\gamma_V)} + \tau \hat{A}_t^{\mathcal{H},\text{GAE}(\lambda_{\mathcal{H}},\gamma_{\mathcal{H}})}. \tag{17}$$

We then train the parameterised policy $\pi_\theta$ using the modified PPO loss function in which the standard advantage is replaced by the soft advantage estimate $\tilde{A}_t^{\pi}$. Additionally, the entropy regularisation term $H(\pi_\theta)$ is replaced by the entropy critic loss $L^{\mathcal{H}}(\omega)$:

$$L_t(\theta, \phi, \omega) = \hat{\mathbb{E}}_t\Big[\min(r_t(\theta)\tilde{A}_t^{\pi}, \text{clip}(r_t(\theta), 1 - \epsilon, 1 + \epsilon)\tilde{A}_t^{\pi})\Big] \tag{18}$$
$$+ c_1(L_t^V(\phi) + c_2 L_t^{\mathcal{H}}(\omega)), \tag{19}$$

where $r_t(\theta)$ is the probability ratio between the behaviour policy $\pi_{\theta_{\text{old}}}$ and the current policy $\pi_\theta$, and $c_1$ and $c_2$ are hyperparameters to be adjusted. The value critic loss $L^V$ is also defined by the mean square error, $L_t^V(\phi) = \hat{\mathbb{E}}_t\left[\frac{1}{2}\left(V(s_t; \phi) - \hat{V}^{\pi}(s_t)\right)^2\right]$ where $\hat{V}^{\pi}$ is the return value estimate.

## 5 Experiments

We empirically show that EAPO supersedes the PPO's entropy regularisation method in various environments. Specifically, EAPO exhibits performance improvement across all 4 MuJoCo (Todorov et al., 2012) and 16 Procgen (Cobbe et al., 2020) benchmark environments over the baseline method. Furthermore, our results highlight the enhanced generalisation capability of the MaxEnt algorithm.

We implemented EAPO using Stable-baseline3 (Raffin et al., 2019) and conducted experiments on environments provided by the Envpool (Weng et al., 2022) library. All empirical results are averaged over runs from 10 random seeds, and we indicate the standard errors. To ensure fair comparisons, a set of common hyperparameters between PPO and EAPO is kept consistent for each MuJoCo and Procgen experiment.

The implementation details and hyperparameters are reported in Appendix B.

### 5.1 Continuous Control Tasks

We measure the performance of EAPO on continuous control tasks in 4 MuJoCo environments (Ant, Walker2d, HalfCheetah and Hopper) against PPO agents with different entropy regularisation coefficients $c \in (0.0001, 0.001, 0.01)$. Moreover, we tested the PPO agent with the setting that the reward is augmented by the entropy reward $-\tau \log \pi(a_t|s_t)$ to evaluate the efficacy of separating the MaxEnt objective. Note that the entropy reward augmented PPO is effectively the same as EAPO with $\gamma_{\mathcal{H}} = \gamma$ and $\lambda_{\mathcal{H}} = \lambda$, and without the entropy critic. We provide the training curves in Figure 1.

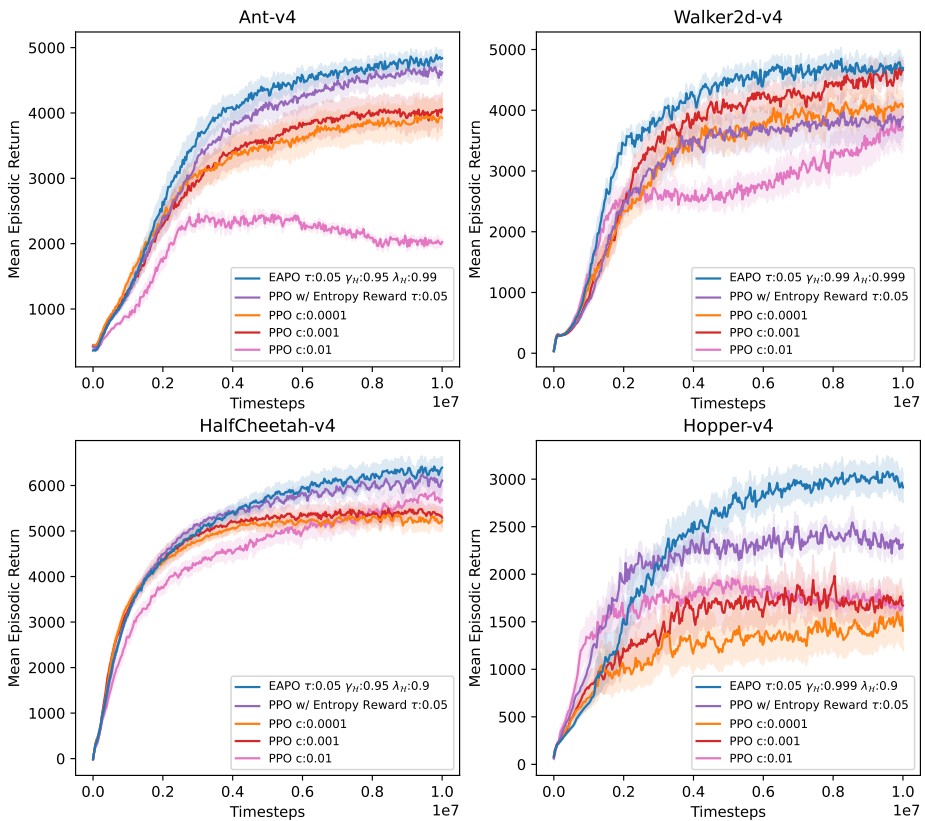

Figure 1: Performance comparison on 4 MuJoCo locomotion tasks. We measured the mean episodic return of the stochastic policy periodically over 100 episodes during the training. Results are averaged from 10 random seeds, and the shaded area indicates the standard error. We compare EAPO to PPO agents with entropy coefficient $c \in (0.0001, 0.001, 0.01)$, and with the entropy reward augmented PPO.

The result shows that EAPO outperforms the conventional entropy regularisation method and the entropy reward augmentation throughout all environments. We found that the best-performing values of $\gamma_{\mathcal{H}}$ and $\lambda_{\mathcal{H}}$ vary depending on the characteristics of the environment, similar to their value counterparts $\gamma$ and $\lambda$, respectively.

Figure 1 also demonstrates that the adjustability adopted by EAPO improves the naive implementation of the MaxEnt policy that augments the entropy reward. In environments where the best $(\gamma_{\mathcal{H}}, \lambda_{\mathcal{H}})$ are close to $(\gamma, \lambda) = (0.99, 0.95)$, (Ant, HalfCheetah) EAPO and the entropy reward augmented PPO performs similarly. However, otherwise, EAPO outperforms it by a wide margin.

## 5.2 PROCGEN BENCHMARK ENVIRONMENTS

We evaluate the generalisation of EAPO on the 16 Procgen benchmark environments (Cobbe et al., 2020). In contrast to MuJoCo tasks, Procgen environments have a discrete action space and provide raw images as observations. Following the evaluation procedure in (Cobbe et al., 2020), we train agents on 200 procedurally generated levels and test the performance as the mean episodic return on 100 unseen levels for each environment. We employ a single set of hyperparameters for all environments. We use the *easy* difficulty setting.

Figure 2 and Table 1 summarise the generalisation test results of EAPO and baseline PPO agents with varying $\tau$ and the entropy bonus coefficient, respectively. EAPO with a lower discount factor of $\gamma_{\mathcal{H}} = 0.8$ and $\lambda_{\mathcal{H}} = 0.95$ surpasses the baseline PPO agent in every environment.

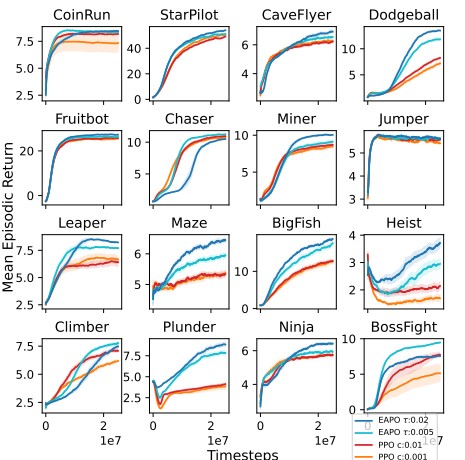
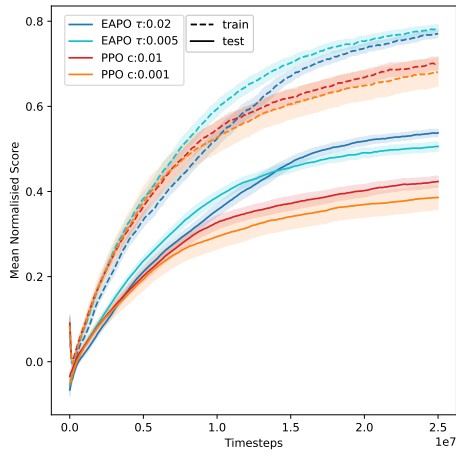

Figure 2: **Left**: Generalisation test results on 16 Procgen (Cobbe et al., 2020) benchmark environments demonstrate EAPO agents with $\tau = 0.02$ and $\tau = 0.005$ against PPO agents with entropy coefficients of $0.001$ and $0.01$. We evaluate agents in 100 levels unseen during the training. EAPO consistently outperforms or at least matches in all environments. Results are averaged over 10 seeds and the shaded area indicates the standard error. **Right**: The mean normalised score for both test and training, computed according to (Cobbe et al., 2020).

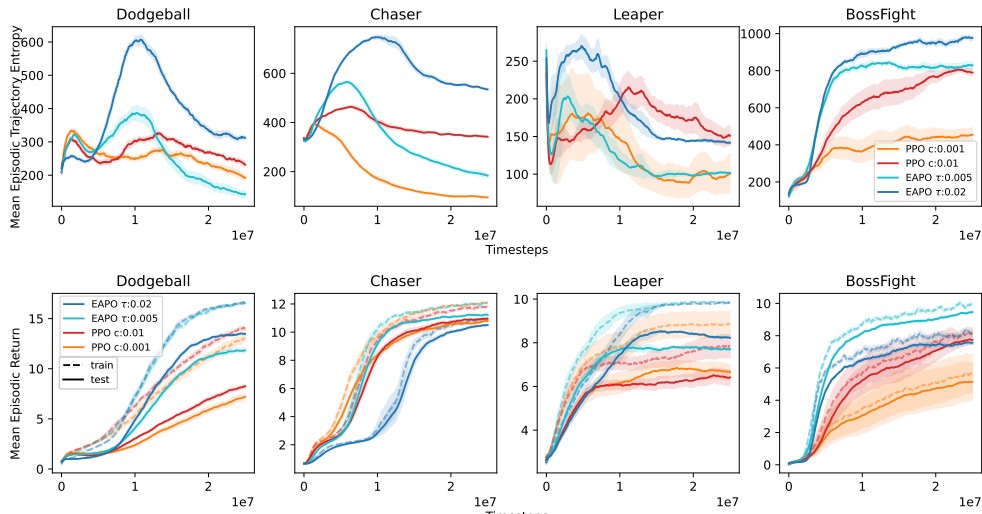

Figure 3: **Top**: Mean episodic trajectory of four selected Procgen environments during the test. The trajectory entropy of an epsiode is calculated as the sum of the negative log probability of the actions of a trajectory using Eq. 5. **Bottom**: Mean episodic return of the selected environments during the test and the training. The higher entropy policy ($\tau = 0.02$) outperforms the lower entropy policy ($\tau = 0.005$) while achieving matching performance during the training (Dodgeball, Leaper) and exhibits a smaller generalisation gap (Dodgeball, Chaser, Leaper).

Figure 2 (Right) shows the improved generalisation capability of high entropy policies. The policy trained with higher temperature $\tau$ favours high entropy trajectories (see Figure 3) and performs similar or worse than the one with lower $\tau$ during the training but achieves better during the test. This result is in coherence with the previous study of (Eysenbach & Levine, 2021) that a MaxEnt policy is robust to the distributional shift in environments.

Table 1: Mean episodic return at the final timestep of tests on 100 unseen levels on 16 Procgen environments. We report the average and the standard deviation from 10 different seeds.

| Enviornment | EAPO | | PPO | |
|---|---|---|---|---|
| | $\tau$:0.02 | $\tau$:0.005 | c:0.01 | c:0.001 |
| CoinRun | **8.34±0.24** | 8.31±0.22 | 8.13±0.22 | 7.38±2.48 |
| StarPilot | **54.33±3.41** | 52.12±1.95 | 49.4±3.72 | 50.57±2.4 |
| CaveFlyer | **7.03±0.32** | 6.48±0.57 | 6.32±0.7 | 6.3±0.63 |
| Dodgeball | **13.64±0.68** | 11.59±0.86 | 8.51±0.98 | 7.2±1.14 |
| Fruitbot | **27.28±0.74** | 26.5±0.9 | 25.91±1.26 | 25.34±1.18 |
| Chaser | 10.5±0.46 | **11.12±0.3** | 11.12±0.45 | 10.64±0.4 |
| Miner | **10.13±0.38** | 8.9±0.84 | 8.6±0.76 | 8.06±0.94 |
| Jumper | 5.7±0.46 | **5.76±0.27** | 5.21±0.45 | 5.17±0.44 |
| Leaper | **8.24±0.52** | 7.75±0.33 | 6.54±0.9 | 6.61±1.06 |
| Maze | **6.5±0.48** | 5.7±0.27 | 5.38±0.56 | 5.45±0.31 |
| BigFish | **19.89±2.14** | 17.88±1.46 | 12.84±1.54 | 12.56±2.86 |
| Heist | **3.75±0.66** | 2.97±0.67 | 2.11±0.54 | 1.68±0.5 |
| Climber | 7.43±0.87 | **8.0±0.5** | 6.78±0.68 | 6.22±0.48 |
| Plunder | **9.0±0.71** | 7.66±1.15 | 4.16±0.55 | 3.94±1.23 |
| Ninja | **6.49±0.47** | 6.07±0.46 | 6.09±0.63 | 5.87±0.58 |
| BossFight | 7.49±0.77 | **9.58±0.66** | 7.61±0.76 | 5.15±3.32 |
| Normalised | **0.54±0.06** | 0.51±0.05 | 0.42±0.07 | 0.38±0.11 |

## 6 CONCLUSION

We have introduced EAPO, a model-free on-policy actor-critic algorithm based on the maximum entropy reinforcement learning framework. Our approach shows that a straightforward extension of existing mechanisms for standard value learning in on-policy actor-critic algorithms to the trajectory entropy objective can facilitate the practical implementation of MaxEnt RL. Through empirical evaluations, our method has been shown to replace the conventional entropy regularisation method and that a more principled entropy maximisation method enhances generalisation. While this paper focuses on PPO, one can seamlessly adapt EAPO to other advantage actor-critic algorithms such as A3C (Mnih et al., 2016) or TRPO (Schulman et al., 2015a). This adaptability lays the groundwork for deeper investigations into the interactions between the MaxEnt RL framework and various components of reinforcement learning algorithms. We anticipate that the inherent simplicity of on-policy algorithms and EAPO will encourage broader applications of the MaxEnt RL algorithm to promising areas like competitive reinforcement learning and robust reinforcement learning.

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

## A  PROOF OF THE SOFT POLICY GRADIENT THEOREM

We begin with the proof of Shi et al. (2019). Let us denote the discounted state distributions induced by the policy $\pi$ as $\rho^\pi(s)$ for the discount factor $\gamma$, and as $\rho^\pi_\mathcal{H}(s)$ for the discount factor $\gamma_\mathcal{H}$, respectively. Then,

$$
\begin{aligned}
\nabla_\theta J(\pi_\theta) &= \mathbb{E}_{\substack{s_0\sim\rho,\\a_t\sim\pi,\\s_{t+1}\sim p}} \left[ (\tilde{Q}^\pi(s_t, a_t) - \tau\log\pi(a_t|s_t) - 1)\nabla_\theta\log\pi(a_t|s_t) \right]\\
&= \sum_s [\rho^\pi(s)[\nabla_\theta\sum_a\pi(s|a)[Q^\pi(s,a) - V^\pi(s)]] + \tau\rho^\pi_\mathcal{H}(s)[\nabla_\theta\sum_a\pi(s|a)(Q^\pi_\mathcal{H}(s,a) - \log\pi(a|s))]]\\
&= \mathbb{E}_{\substack{s_0\sim\rho,\\a_t\sim\pi,\\s_{t+1}\sim p}} [\nabla_\theta\log\pi_\theta(a_t|s_t)A^\pi(s_t,a_t)] + \tau\sum_s\rho^\pi_\mathcal{H}(s)[\nabla_\theta\sum_a\pi(s|a)(Q^\pi_\mathcal{H}(s,a) - V^\pi_\mathcal{H}(s))]\\
&= \mathbb{E}_{\substack{s_0\sim\rho,\\a_t\sim\pi,\\s_{t+1}\sim p}} [\nabla_\theta\log\pi_\theta(a_t|s_t)A^\pi(s_t,a_t)] + \tau\mathbb{E}_{\substack{s_0\sim\rho,\\a_t\sim\pi,\\s_{t+1}\sim p}} [\nabla_\theta\log\pi_\theta(a_t|s_t)A^\pi_\mathcal{H}(s_t,a_t)]\\
&= \mathbb{E}_{\substack{s_0\sim\rho,\\a_t\sim\pi,\\s_{t+1}\sim p}} \left[ \nabla_\theta\log\pi_\theta(a_t|s_t)\tilde{A}^\pi_t(s_t,a_t) \right]. \qquad\qquad\qquad \square
\end{aligned}
$$

## B  HYPERPARAMETERS AND IMPLEMENTATION DETAILS

### B.1  HYPERPARAMETERS

We use the default values in stable-baseline3 (Raffin et al., 2019) and envpool (Weng et al., 2022) libraries for the settings not specified in the table 2. EAPO-specific hyperparameters are reported in Table 3. The parameters for the MuJoCo tasks are found in a coarse hyperparameter search.

### B.2  NETWORK ARCHITECTURE

For MuJoCo tasks, we used simple ReLU networks for the policy and the critics with hidden layers of depth [256, 256] and [512, 512], respectively. We implemented the entropy critic as the independent output layer that shares the hidden layers with the value network. We use the state- and action-dependent $\sigma$ output for the Gaussian policy with the Softplus output layer.

In Procgen benchmark experiments, we adopt the same IMPALA CNN architecture used in Cobbe et al. (2020). The entropy critic is implemented analogously to the conventional value network.

Table 2: Common PPO hyperparameters.

| Parameter | MuJoCo | Procgen |
|---|---|---|
| Timesteps | 10M | 25M |
| Num. Envs | 32 | 64 |
| Num. Steps | 256 | 256 |
| Learning Rate | $5 \times 10^{-5}$ | $5 \times 10^{-4}$ |
| Batch Size | 1024 | 2048 |
| Epochs | 10 | 3 |
| Discount Factor $\gamma$ | 0.99 | 0.995 |
| GAE $\lambda$ | 0.95 | 0.8 |
| Clip Range $\epsilon$ | 0.15 | 0.1 |
| Max Grad. Norm. | 1.0 | 0.5 |
| $\log \sigma$ Init. | -1 | None |
| Minibatch Advantage Norm. | True | True |
| Observation Norm. | True | False |
| Reward Norm. | False | False |
| PopArt $\beta$ | 0.03 | 0.03 |
| Value Loss Coeff. | 0.5 | 0.5 |
| Entropy Coeff. | (0.01, 0.001, 0.0001) | (0.01, 0.001) |

Table 3: EAPO specific hyperparameters.

| Parameter | MuJoCo | Procgen |
|---|---|---|
| $\gamma_{\mathcal{H}}$ | 0.95 (Ant) 0.99 (Walker2d) 0.95 (HalfCheetah) 0.999 (Hopper) | 0.8 |
| $\lambda_{\mathcal{H}}$ | 0.99 (Ant) 0.999 (Walker2d) 0.9 (HalfCheetah) 0.9 (Hopper) | 0.95 |
| EAPO $c_1$ | 0.5 | 0.5 |
| EAPO $c_2$ | 0.1 | 0.5 |
| $\tau$ | 0.05 | (0.005, 0.02) |

