# OpenReview forum: "Maximum Entropy On-Policy Actor-Critic via Entropy Advantage Estimation"
_ICLR.cc/2024/Conference — ICLR 2024 Conference Withdrawn Submission_

### Official Review · Reviewer_62Vd · 2023-10-28

**Soundness:** 2 fair
**Presentation:** 2 fair
**Contribution:** 2 fair
**Rating:** 3
**Confidence:** 4

**Summary:**

This paper studies the MaxEnt RL framework under on-policy actor-critic context, and proposes the Entropy Advantage Policy Optimisation (EAPO) algorithm. The basic idea is to independently estimate the value advantage function and the entropy advantage
function, and then combine them to derive the overall soft advantage function. The empirical results are conducted on 4 MuJoCo benchmarks and 16 Procgen benchmarks, against entropy-regularized versions of PPO.

**Strengths:**

1. Extending MaxEnt RL to on-policy algorithms seems to be a reasonable idea.

2. The EAPO is heuristic but makes sense.

3. EAPO is empirically shown to outperform variants of PPO with entropy regularization.

**Weaknesses:**

1. It is not 100% clear to me why MaxEnt RL should be studied on the on-policy context. The paper motivates it simply by saying some tasks are more suited for on-policy RL, with some superfacial examples at the end of Introduction, but it fails to show the actual advantage of on-policy MaxEnt RL over off-policy MaxEnt RL from either a theoretical or an empirical point of view.

2. The key technical contribution is the separate advantage computation for the value and entropy, which seems to be a heuristic and the novelty is quite limited.

3. Unfortunately, the empirical performance is not convincing, either.
- It only compares to (variants of) PPO, whose performance is known to be inferior to SAC in MuJoCo benchmarks. If the paper claims that on-policy MaxEnt RL is better in some tasks, the best way to empirically show it is to conduct experiments on problems where it does outperform off-policy RL. Otherwise, the claim appears to be weak and not well grounded.
- The paper claims that on-policy RL is better at generalization and shows experiments on the Procgen benchmarks. But if on-policy RL is better than off-policy RL in generalization, comparison against off-policy RL algorithms is necessary to make such a claim.

4. The writing needs to be improved. E.g.,
- In Introduction, the paper uses Zhang et al. 2018 to argue that data overfitting compromises generalization. But Zhang et al. 2018 did not say that off-policy RL is more prone to be overfitting. Better justificaiton should be provided.
- The paper puts down the Soft Policy Gradient theorem without referencing it to Shi et al. 2019 in the main paper. It appears to be the contribution of this paper, but it really is a very simple extension of that in Shi et al. 2019. This should be made clearer.

**Questions:**

N/A. See my concerns above.

---

### Official Review · Reviewer_ztEn · 2023-10-29

**Soundness:** 2 fair
**Presentation:** 1 poor
**Contribution:** 2 fair
**Rating:** 3
**Confidence:** 4

**Summary:**

This paper replaces the entropy term in PPO with a trajectory entropy term which is common used in MaxEnt RL. And following experimental results show this trick would improve performance.

**Strengths:**

Change the type of entropy term might help performance pratically.

**Weaknesses:**

This paper is more like a proposal or an experimental report, but not a done paper.

1. Poor writing and limited literature review in Introduction and Related Works.

2. Proposing a practical trick is ok for a paper, but you must provide abundant theoretical and empirical support.

   * Lack of theoretical support for the proposed trick.

   * Marginal performance improvement compared to PPO in MuJoCo tasks.

   * Only implement the trick on PPO but claim its generalizability.

   * Few results on popular standard benchmarks.

**Questions:**

**Suggestions**

Just having a TRICK and a handful of experiments is not at all sufficient; my suggestions are:

1. give theoretical support

2. add experiments on DMControl, Meta-World.

3. implement the trick on other on-policy actor-critic algorithms and test it.

4. investigate the trick on some off-policy actor-critic algorithms

5. survey more related literature and rewrite the introduction and related works.



I recommend some papers on actor-critic algorithms and entropy term designs:

**Entropy-term designs**

* Elad Hazan, Sham Kakade, Karan Singh, and Abby Van Soest. Provably efficient maximum entropy exploration. In International Conference on Machine Learning, 2019.
* Riashat Islam, Zafarali Ahmed, and Doina Precup. Marginalized state distribution entropy regularization in policy optimization. arXiv preprint arXiv:1912.05128, 2019.
* Seungyul Han and Youngchul Sung. Diversity actor-critic: Sample-aware entropy regularization for sample-efficient exploration. In International Conference on Machine Learning, 2021.

**Actor-Critic Algorithms**

* Han S, Sung Y. A max-min entropy framework for reinforcement learning. Advances in Neural Information Processing Systems, 2021.
* Hao Sun, Lei Han, Rui Yang, Xiaoteng Ma, Jian Guo, and Bolei Zhou. Optimistic curiosity exploration and conservative exploitation with linear reward shaping. In Advances in Neural Information Processing Systems, 2022.

* Ji, Tianying, et al. Seizing Serendipity: Exploiting the Value of Past Success in Off-Policy Actor-Critic. arXiv preprint arXiv:2306.02865, 2023.

---

### Official Review · Reviewer_oWp6 · 2023-10-30

**Soundness:** 2 fair
**Presentation:** 3 good
**Contribution:** 2 fair
**Rating:** 3
**Confidence:** 4

**Summary:**

Entropy regularization is commonly used in on-policy policy gradient methods like PPO to improve performance and stability. However, because entropy regularization only maximizes policy entropy in visited states, it has led to the development of Maximum Entropy Reinforcement Learning (MaxEnt RL), which aims to maximize the cumulative reward and the entropy of trajectories. Algorithms such as Soft Actor-Critic (SAC) have been extensively researched in the off-policy RL, but there is a lack of work for on-policy methods. In this paper, the authors propose an on-policy actor-critic algorithm called EAPO within the MaxEnt RL framework. This algorithm separates the entropy objective from the MaxEnt RL objective and introduces an entropy critic. This separation enables finer control over the entropy term and makes it easier to adjust the balance between the entropy term and the value term. Through experiments conducted in Mujoco and Procgen environments, EAPO demonstrates improved performance compared to PPO.

**Strengths:**

1. There have been many approaches to improving the performance of on-policy algorithms by using entropy regularization to increase exploration and enhance stability. There has also been a significant body of research on using the MaxEnt RL framework, which increases the expected trajectory entropy in off-policy algorithms like SAC to improve performance. However, the use of MaxEnt RL in on-policy algorithms has been underexplored. Through research that combines PPO with MaxEnt RL, a new algorithm was developed, and experimental results demonstrated better performance compared to PPO.
2. By combining the on-policy RL algorithm PPO with the MaxEnt RL framework, both stability and performance were enhanced. Moreover, separating the entropy objective from the estimate has several advantages. Firstly, the entropy term can have a distinct discount factor, allowing for effective reduction of the horizon, addressing the long trajectory issue present in traditional MaxEnt RL. Additionally, by using Generalized Advantage Estimation (GAE) for the entropy term, the trade-off between variance and bias can be controlled. Separating the objective function and introducing a new entropy critic enables independent optimization of the entropy term while fine-tuning control.
3. Off-policy algorithms may suffer from data overfitting due to the reuse of learning data and a lack of generalization in unseen environments. By merging MaxEnt RL, which is robust to distributional shifts in environments, with on-policy RL, generalization was improved. Experiments showed enhanced performance in unseen environments. Furthermore, the inherent simplicity of on-policy RL and EAPO, along with the generalization property of MaxEnt RL, open up new avenues for applications in competitive and robust RL fields.

**Weaknesses:**

1. Approaches like the soft advantage function and GAE have been explored extensively in previous off-policy MaxEnt RL research, and there have been studies combining on-policy RL with the MaxEnt RL framework. Since the performance difference between simply adding entropy reward to PPO and the proposed algorithm appears to be similar in the experiments, additional contributions would be valuable.
2. The direction of integrating MaxEnt RL into PPO, which traditionally used entropy regularization alone, is promising. It would be even more motivating to highlight the advantages that arise from combining MaxEnt RL with on-policy algorithms in this paper. The paper could benefit from demonstrating the inherent simplicity and lack of overfitting in unseen environments with experiments, both strengths of on-policy algorithms in this paper. For example, a comparison with off-policy RL algorithms in the Procgen unseen environment would be beneficial. Additionally, performance experiments in the MuJoCo environment would be good not only to compare PPO, but also to off-policy algorithms and actor-critic algorithms.
3. The paper's contribution of separating the entropy objective from the MaxEnt RL objective and proposing a new entropy critic is valuable. It would be advantageous to demonstrate experimentally whether the associated benefits materialize. While it is possible to fine-tune control with additional discount factors and hyperparameters in the experiments, it would be beneficial to include an ablation study. Additionally, demonstrating the reduction in horizon as an advantage through experiments would strengthen the paper's findings. Furthermore, showing the decrease in variance in the entropy term due to the use of GAE in experiments would increase the validity.

**Questions:**

1. Although an entropy critic was added to optimise the entropy objective function separately, the experiments utilized parameter sharing with the existing value critic. This approach offers the advantage of reducing computational costs. However, it appears that using this method is similar to not adding a new entropy critic and instead incorporating the final layer into the existing value critic, then separating and training only the objective function. I'm also curious about the performance difference when parameter sharing is not used compared to when it is used.
2. In this paper, it was mentioned that EAPO's theoretical guarantees are unclear in a regularised MDP setting. I'm curious if adding the MaxEnt RL framework independently to the on-policy policy gradient theoretically does not hinder convergence. Additionally, if we were to use TRPO (Trust Region Policy Optimisation) instead of PPO as the baseline in the setting proposed in this paper, could we potentially establish theoretical guarantees?
3. In the experiments with MuJoCo environments, it appears that entropy reward augmented PPO is an interpretation of adding entropy rewards to the existing PPO. I'm curious whether entropy reward augmented PPO kept the entropy regularisation intact or removed it and simply added entropy rewards. Additionally, I would like to know your thoughts on using both entropy regularization and the maximum entropy RL framework simultaneously.

---

### Official Review · Reviewer_273E · 2023-10-30

**Soundness:** 3 good
**Presentation:** 2 fair
**Contribution:** 2 fair
**Rating:** 3
**Confidence:** 3

**Summary:**

The authors presents a novel method of on-policy actor-critic algorithm based on entropy regularization with separated critics for reward and entropy parts of regularized value. This separation allows to use separate discount factors to provide more proper balance between value and MaxEnt objectives. Finally, authors extend Proximal Policy Optimization into MaxEnt paradigm and verify their ideas in 4 MuJoCo and 16 Procgen environments.

**Strengths:**

- Novel idea with separated critics that allows to deal with the problem of different scaling of rewards and policy entropy;
- One of the first algorithms in the nearly empty niche of MaxEnt on-policy RL algorithms;

**Weaknesses:**

The major weakness of the presented paper is weak empirical validation of the proposed method: lack of baseline comparisons and ablation studies.

- No experimental comparison to other on-policy methods such as TRPO and Mirror Descent Policy optimization (MDPO);
- Lack of ablations study: for example, the effect of different discounting and GAE coefficients for reward and entropy value is not studied properly;

Tomar, M., Shani, L., Efroni, Y., & Ghavamzadeh, M. (2020). Mirror descent policy optimization. ICLR 2022

**Questions:**

- Is it possible to introduce additional KL regularization to this method, akin to MDPO or Munchausen DQN?
- Is the proposed method less sensitive to hyperparameter choice such a learning rate, compared to PPO?

Vieillard, N., Pietquin, O., & Geist, M. (2020). Munchausen reinforcement learning. *Advances in Neural Information Processing Systems*, *33*, 4235-4246.